# Temporal measures of oropharyngeal swallowing events identified using ultrasound imaging in healthy young adults

Elaine Kwong[1,2]*, Phoebe Tsz-Ching Shek[1], Man-Tak Leung[1], Yong-Ping Zheng[2,3], Wilson Yiu Shun Lam[1,2]

1 Department of Chinese and Bilingual Studies, The Hong Kong Polytechnic University, Hung Hom, Hong Kong, 2 Research Institute for Smart Ageing, The Hong Kong Polytechnic University, Hung Hom, Hong Kong, 3 Department of Biomedical Engineering, The Hong Kong Polytechnic University, Hung Hom, Hong Kong

* elaine-yl.kwong@polyu.edu.hk

**Data Availability Statement:** All relevant data are within the paper and its Supporting Information files.

## Abstract

Swallowing is a complex process that involves precise coordination among oral and pharyngeal structures, which is essential to smooth transition of bolus and adequate airway protection. Tongue base retraction and hyolaryngeal excursion are two significant swallowing movements, and their related events can be examined using ultrasound imaging, which is physically and radioactively non-invasive. The present study aimed to 1) establish the temporal sequences and timing of swallowing events identified using ultrasound imaging, and 2) investigate the variability of the above temporal sequences and 3) investigate the effect of bolus type on the variability of temporal sequences in non-dysphagic individuals. Forty-one non-dysphagic young adults of both genders (19 males and 22 females) participated in the study. Ultrasound images were acquired mid-saggitally at their submental region during swallowing of boluses with different volume (i.e. 5mL or 10mL) and consistencies (i.e. IDDSI Levels 0 and 4). Timing and sequence of six events 1) displacement onset (*TBOn*), 2) maximum displacement (*TBMax*) and 3) displacement offset of tongue base retraction (*TBOff*); and, 4) displacement onset (*HBOn*), 5) maximum displacement (*HBMax*) and 6) displacement offset of the hyoid bone excursion (*HBOff*) were extracted from the ultrasound images. Out of the 161 swallows, 85.7% follow a general sequence of *HBOn < TBOn < HBMax < TBMax < HBOff < TBOff*. Percentage adherence to six anticipated paired-event sequences was studied. Results suggested the presence of individual variability as adherence ranged from 75.8% to 98.1% in four of the anticipated sequences, leaving only two sequences (*HBOn < TBMax* and *TBMax < HBOff*) obligatory (i.e. 100% adherence). For non-obligatory sequences, it was found that bolus type may have an effect on the level of adherence. Findings of the present study lay the groundwork for future studies on swallowing using ultrasound imaging and also the clinical application of ultrasonography.

**Funding:** E.K., M.-T. L. and Y.-P. Z. received the Departmental General Research Fund (P0030029) of the Department of Chinese and Bilingual Studies, The Hong Kong Polytechnic University (https://www.polyu.edu.hk/cbs/web/en/). The funders had no role in study design, data collection and analysis, decision to publish, or preparation of the manuscript.

**Competing interests:** The authors have declared that no competing interests exist.

## Introduction

Swallowing is a complex process that involves coordination among over 30 muscles and their respective cranial nerves to achieve transition of bolus as well as airway protection at the same time [1]. Precise coordination is essential to safe swallows and disruptions to it may result in dysphagia or swallowing disorders, which may in turn lead to major health conditions like aspiration pneumonia, dehydration, and malnutrition.

The process of swallowing is commonly divided into four stages; namely, 1) the oral preparatory state which involves mostly mastication, 2) the oral stage which involves propulsion of the processed bolus towards the oropharynx, 3) the pharyngeal stage which starts when the bolus reaches the anterior faucial arch and ends when the bolus has entirely passed the upper esophageal sphincter (UES), and 4) the esophageal stage in which the bolus travels along the esophagus before reaching the lower esophageal sphincter [2, 3]. While traditionally swallowing was perceived as a process with temporal stages demarcated by the four stages, evidence has shown that there is temporal overlapping of the physiological events categorized in these stages [4]. A sequential pattern, however, can be observed by examining the details of the events. McConnel and his colleagues' study [5] using synchronized fluoroscopy and manometry investigated the sequential pattern of the laryngeal movement preceding the tongue base movements. In Mendell and Logemann's study [6], the onset of laryngeal movement and hyoid movement was also observed to precede the onset of tongue base movement consistently. Disruptions of the timing of these swallowing events were found to be related to the dysfunctions of swallowing mechanisms. In Lee and his colleagues' study [7] with a group of patients with dysphagia, it was observed that patients who aspirate on thin liquid showed significant delay in rising time and peak of laryngeal movement relative to the start of pharyngeal swallow. Head and neck cancer patients who had undergone radiotherapy were observed to have delayed hyoid bone movement relative to pharyngeal bolus transit, yet it was compensated by an earlier arrival of maximum hyoid bone movement [8].

To date, most of the studies on the temporal aspects of swallowing events adopt videofluoroscopy (e.g. [8, 9]), while others adopt manometry or both (e.g. [10, 11]). Videofluoroscopy and manometry are, nevertheless, either physically or radioactively invasive; thus the number of trials on subjects should be kept minimal. Ultrasonography, on the other hand, has the advantages of being non-invasive and radiation-free. Previous studies showed that ultrasound imaging allows the observation of oropharyngeal movements when the transducer is placed at the submental position [12, 13]. This technique was also found to have higher sensitivity in assessing the oral phase of patients with degenerative diseases as compared to videofluoroscopy [14]. Hyoid bone movement was found to be reliably observed using ultrasound imaging [13, 15, 16]. In contrast to most of the abovementioned studies, which acquired ultrasound images with a convex-shaped transducer placed on the midsagittal plane of the submental region, Matsuo and colleagues used a linear-shaped transducer that placed on a plane slightly deviated from the midsagittal one. In addition to measuring the displacements of the hyoid bone and larynx, they also tried to compute the ratio between the two displacements to give an index of coordination between the geniohyoid and thyrohyoid muscles [17]. In relation to the temporal aspects of swallowing, Stone and Shawker [18] investigated the timing relationship between posterior tongue movement and hyoid bone movement. The adoption of ultrasonography over videofluoroscopy or manometry is hence promising.

Two specific swallowing movements, namely, tongue base retraction and hyoid bone excursion were of interest in this study. Besides being readily identifiable, precise temporal coordination of these movements are significant in swallowing safety in the pharyngeal stage based on the traditional Four-stage model. Tongue base retraction is an important action for the

delivery of bolus to the pharynx. Hyoid bone excursion 1) denotes the superior-anterior move-ment of the larynx and facilitates the posterior tilting of the epiglottis, 2) widens the pharyngeal area and creates a suction force for the bolus towards the esophagus, and 3) promotes the opening of the UES and entrance of bolus into the esophagus [2]. Further, it was shown that patients with swallowing dysfunction may have compensatory mechanisms that safeguard air-way protection, for instance, reaching the maximum displacement point earlier [8]. The pres-ent study did not only measure the onset time of tongue base retraction and hyoid bone excursion, but also the time at which maximum displacements are reached and the offset time of the two movements are also measured and compared. It was hoped that a comprehensive temporal profile of swallowing events related to the two movements would be established.

With respect to the temporal aspect of swallowing events, variability has been reported in the literature. Kendall and colleagues [19] identified variability in event sequences and found only four obligatory sequences in paired-events out of the 12 paired-events studied. Molfenter and colleagues [20] failed to replicate the obligatory sequences and found even greater extent of variability in the healthy subjects. A review by Molfenter and Steele [21] on 46 published studies suggested different sources of variability, with bolus property being one of them, in the temporal measures of swallowing. Studies by Nagy et al. [22] and Nagy et al. [23] found that bolus volume and bolus consistency respectively may alter velocity and displacement of hyoid bone movement, resulting in difference in timing of swallowing events. These further suggest that variability in the events associated with tongue base retraction and hyoid bone excursion, as well as the effects of bolus types on such variability shall be investigated.

The present study aimed to 1) establish the temporal sequences and timing of swallowing events identified using ultrasound imaging, 2) investigate the variability of the above temporal sequences in non-dysphagic individuals, and 3) investigate the effect of bolus type on the vari-ability of temporal sequences in non-dysphagic individuals. To the best of the authors' knowl-edge, this study is the first to investigate swallowing events, and also their variability, related to tongue base retraction and hyoid bone excursion collectively using ultrasonography. Effects of bolus property on swallowing events captured on ultrasound images had not been previously reported in the literature. Further, this study did not only focus on the onset of the two swal-lowing movements, but also their maximum displacement and offset as swallowing events. Findings of this study would lay the foundation for future studies on swallowing kinematics using ultrasound imaging. The normative temporal data obtained would also allow future comparisons with dysphagic individuals.

## Materials and methods

### Participants

The present study was approved by the Human Subject Ethics Sub-committee, the Hong Kong Polytechnic University (Ref. HSEARS20190521007). Written informed consent was obtained from all participants. Forty-one adults (19 males and 22 females) who aged from 20 to 30 years participated in the present study. All participants should have no structural malformation of the oral cavity, history of dysphagia and have not received any surgery and/or medication that may have an effect on swallowing functions.

### Materials and equipment

Ultrasound images were acquired using the Aixplorer® Multiwave™ Ultrasound System with a XC6-1 convex transducer. Action® BOL-X-I gel pads with film (dimension: 10 cm x 10 cm x 1 cm) were used to ensure proper fitting of the transducer to the submental area of sub-jects. Some subjects required an additional strip of gel pad (dimension: 10 cm x 2 cm x 1 cm)

to eliminate air gaps. Gum-based thickener (Neo-High Toromeal III, FoodCare) was used to produce water boluses at the desired consistency levels.

## Procedures

**Ultrasound image acquisition.** Subjects sat comfortably in an upright position with their head supported against a wall or with manual support by the examiner throughout the image acquisition process. The ultrasound transducer was placed on the mid-sagittal plane in the submental area of subjects. Readers may refer to figures of a previous study [24] for the placement and fixation of gel pads and the transducer. All ultrasound images were acquired by two final-year Master of Speech Therapy students who were trained to manage dysphagia (one of them is the second author P.S.). Before the commencement of image acquisition, both examiners were trained by a research personnel, who had at least three years of experience in conducting ultrasonographic data collection, on the operation of the ultrasound system and had gone through at least ten hours of practice on a number of individuals. Water boluses of different volumes and at different consistencies based on the International Dysphagia Diet Standardisation Initiative (IDDSI) framework [25] were prepared. Ultrasound images were recorded at a fixed frame rate of 32 frames per second when subjects were instructed to swallow the following bolus types in random orders:

1. *5mL x thin*: 5mL water bolus at IDDSI Level 0;

2. *5mL x thick*: 5mL water bolus at IDDSI Level 4;

3. *10mL x thin*: 10mL water bolus at IDDSI Level 0; and,

4. *10mL x thick*: 10mL water bolus at IDDSI Level 4.

These bolus types were selected as exemplars of small and large boluses with thin and thick consistencies and to elicit possible variabilities in swallowing kinematics, if any. For each bolus type, three trials were undergone by each subject and ultrasound images of the respective trial were recorded. In each trial, subjects were fed the designated bolus using a syringe. They were required to hold the bolus in the anterior part of the tongue surface before being instructed to swallow by the examiners. All recorded trials were reviewed by the second author P.S. on image quality based on clarity and artifacts. Out of the three recorded trials, only the one with the best image quality would be selected for the subsequent image processing and data extraction. One female subject and two female subjects failed to swallow the *5mL x thick* and *10mL x thick* boluses respectively. Data sets of these bolus types, therefore, only consisted of data from 40 and 39 subjects respectively. A sample ultrasound image of the relevant structures is shown in Fig 1. The contour of the tongue surface and tongue base are pointed by green arrows (A), the position of hyoid bone is annotated as the red dot with the red line indicating the acoustic shadow margin of the hyoid bone (B), and the geniohyoid muscle is segmented as the yellow polygon (C).

**Data extraction.** Ultrasound images acquired were examined frame-by-frame by the second author P.S. The frames at which the following swallowing events had occurred were extracted:

1. displacement onset, maximum displacement and displacement offset of tongue base retraction (i.e. *TBOn*, *TBMax* and *TBOff* respectively. See Fig 2 for illustration.); and,

2. displacement onset, maximum displacement and displacement offset of the hyoid bone excursion (i.e. *HBOn*, *HBMax* and *HBOff* respectively. See Fig 3 for illustration).

"Displacement onset" was defined as the point at which the corresponding structure started to leave its at-rest position and commence movement. "Maximum displacement" was defined

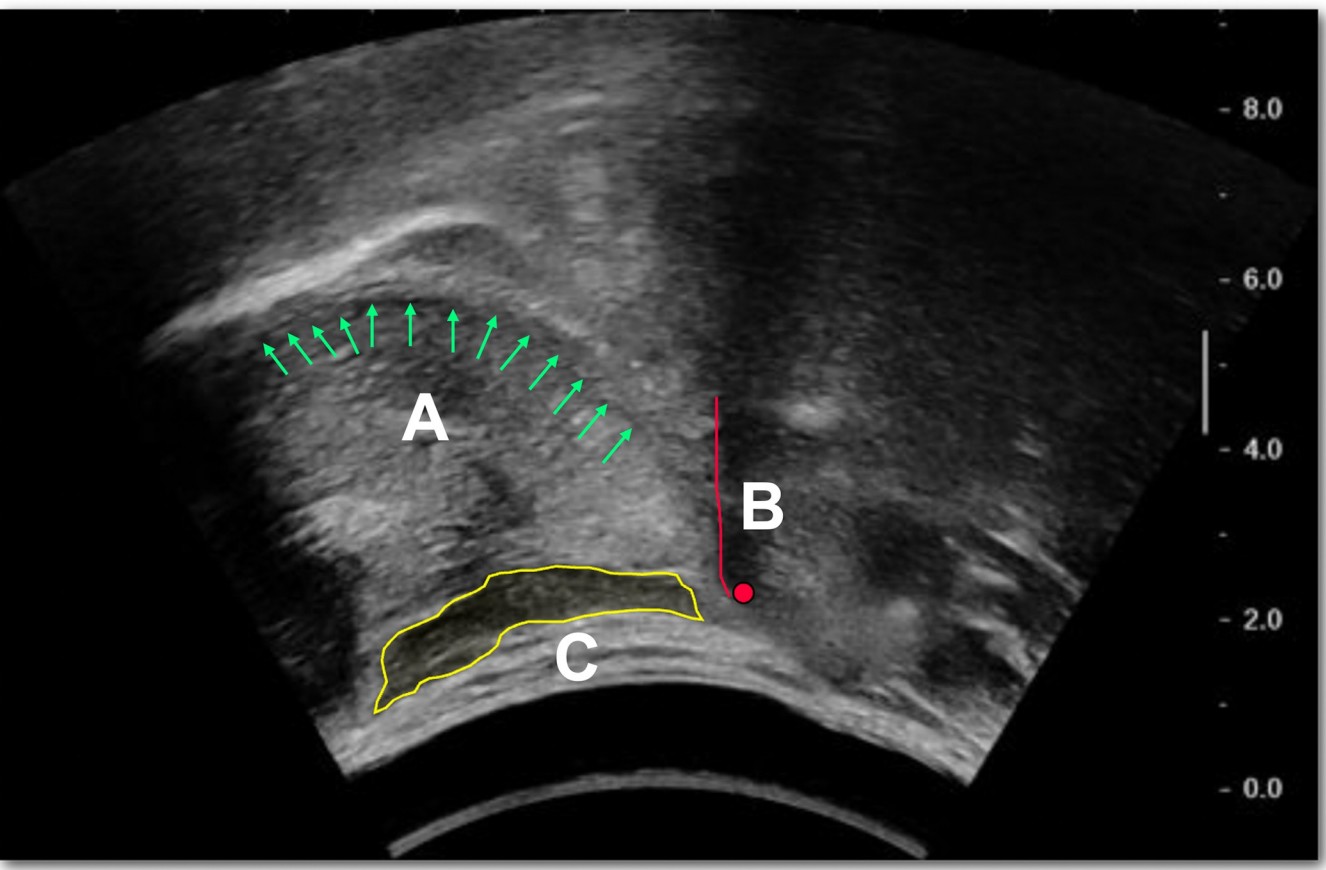

**Fig 1.** Sample ultrasound image showing the tongue surface and tongue base (A), hyoid bone (B), and geniohyoid muscle (C).

as the point at which the corresponding structure reached the position farthest away from its at-rest position. "Displacement offset" was defined as the point at which the corresponding structure started to return to its at-rest position from the maximally displaced position. The timing data of events were extracted by converting the differences in frame number to the time differences (in ms) between the reference event, *HBOn*, and the corresponding event. Fifteen percent of the ultrasound images were selected randomly and data extraction was repeated by the examiner for intra-examiner reliability. Another 30% of all the acquired images were selected randomly and data were extracted from them by an independent examiner (also a final-year Master of Speech Therapy students who were trained to manage dysphagia) for inter-examiner reliability.

## Statistical analyses

All statistical analyses were conducted using the IBM SPSS Statistics 25 software. Intra-examiner reliability was analyzed using Intra-class Correlations Coefficient (ICC) based on a single measure, absolute agreement, two-way mixed-effects model Inter-examiner reliability was analyzed using ICC based on a single rater, absolute agreement, two-way mixed-effects model. The effects of bolus volume and consistency on the timing of swallowing events, as measured by the time differences between the reference event and the corresponding events, were analysed using two-way ANOVAs. The differences in level of adherence between different bolus types were analysed using Pearson's Chi-squared tests or Fisher's exact tests, depending on

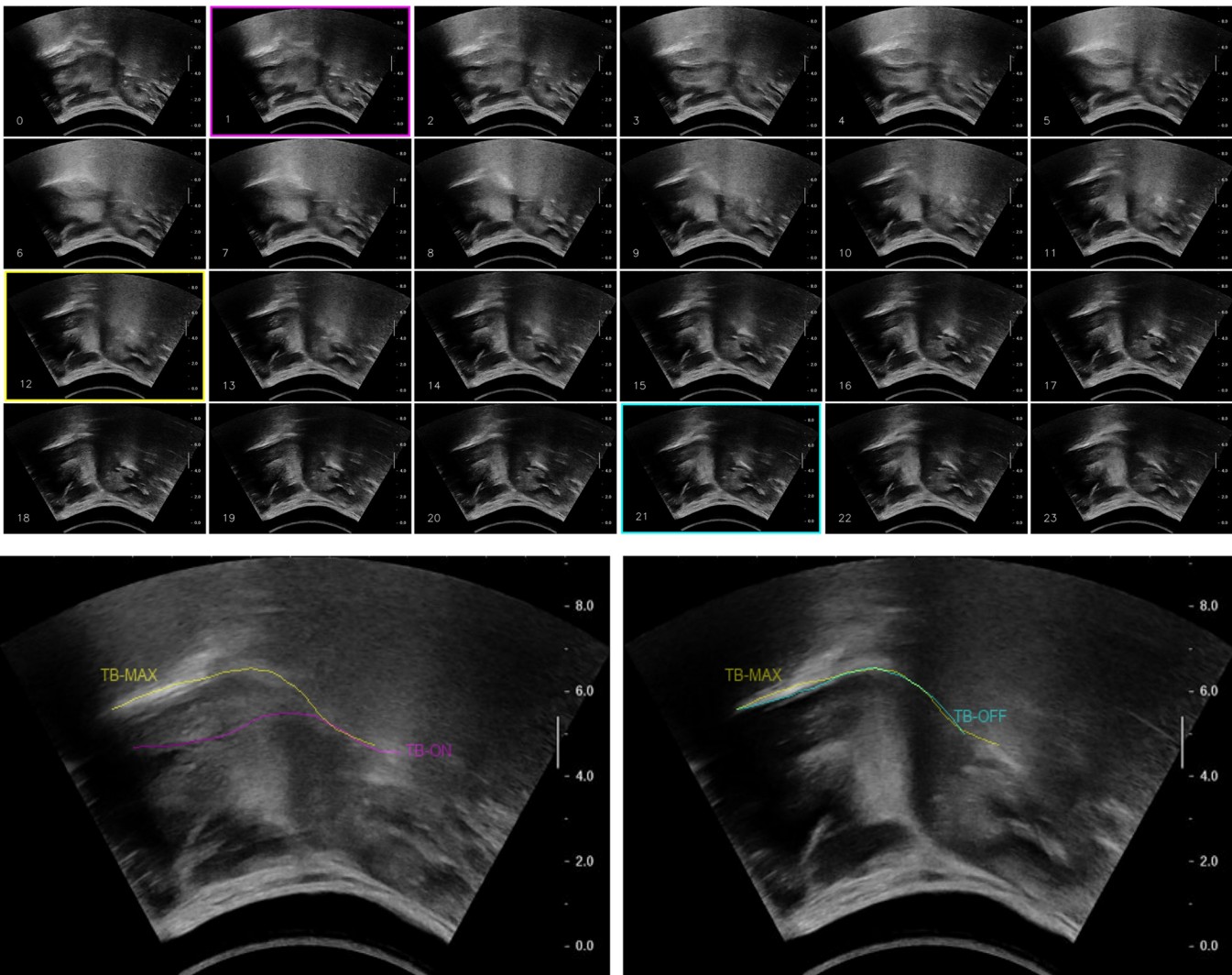

**Fig 2.** (a) Numbered sequential juxtaposition of sample tongue base retraction frames (from left to right, top to bottom); (b) corresponding superimpositions of *TBOn/TBMax* (left) and *TBMax/TBOff* (right). *Note*: Magenta frame = *TBOn*, Yellow frame = *TBMax*, Cyan frame = *TBOff*.

whether the condition of having an expected count of five or more in all cells was met [26]. The association between bolus type and adherence to anticipated sequences of paired-events was analyzed using Fisher's exact tests.

## Results

Table 1 summarizes the results of intra- and inter-examiner reliabilities in swallowing event identification. Excellent intra-examiner reliability and moderate inter-examiner reliability were noted for all swallowing events identified [27].

### Temporal sequence of swallowing events related to tongue base retraction and hyoid bone excursion

The mean time differences between the reference event (i.e. *HBOn*) and other swallowing events are summarized in Table 2 and illustrated in Fig 4. Among all the events being

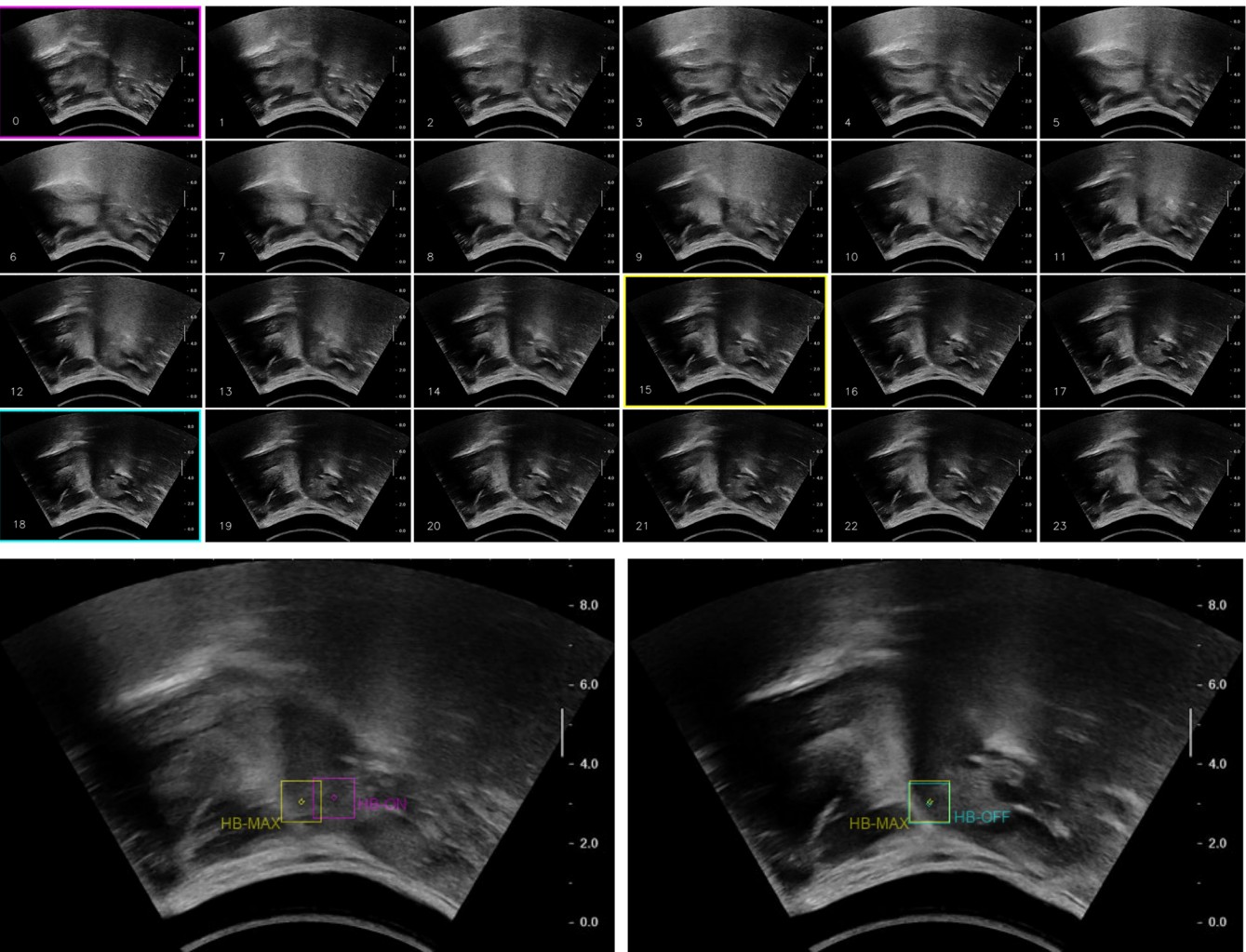

**Fig 3.** (a) Numbered sequential juxtaposition of sample hyoid bone excursion frames (from left to right, top to bottom); (b) corresponding superimpositions of *HBOn/HBMax* (left) and *HBMax/HBOff* (right). *Note*: Magenta frame = *HBOn*, Yellow frame = *HBMax*, Cyan frame = *HBOff*.

studied, *HBOn* was consistently found to be the first swallowing event to occur, regardless of bolus volume and consistency. This finding was in line with the four-stage model of swallowing suggested by Pongpipatpaiboon and colleagues [2]. The sequence *HBOn < TBOn < HBMax < TBMax < HBOff < TBOff* is generally followed in swallows of all bolus types. The percentage of swallowing trials that adhered to the sequence above was found to be 85.7% (138/161). The percentage adherence when swallowing different bolus types ranged from 84.6% (33/39, for *10mL x thick* boluses) to 87.5% (35/40, for *5mL x thick* boluses) (See Fig 5).

Table 3 shows the results of two-way ANOVAs on the time difference between *HBOn* and the corresponding events. All interaction effects and main effects of bolus volume were statistically non-significant. On the contrary, main effects of bolus consistency for all events were significant ($p \leq 0.033$), with the exception of *TBOn* ($F(1, 157) = 2.674$, $p = 0.104$, $\eta_p^2 = 0.017$). Effect sizes for the main effect of bolus consistency were small for *TBMax* ($\eta_p^2 = 0.029$) and *TBOff* ($\eta_p^2 = 0.033$) and medium for *HBMax* ($\eta_p^2 = 0.066$) and *HBOff* ($\eta_p^2 = 0.065$).

**Table 1. Intra- and inter-examiner reliabilities in swallowing event identification.**

| Swallowing event | Intra-examiner reliability | | Inter-examiner reliability | |
|---|---|---|---|---|
| | ICC | *p* | ICC | *p* |
| HBOn | 1.000 | < 0.001 | 0.728 | < 0.001 |
| HBMax | 1.000 | < 0.001 | 0.727 | < 0.001 |
| HBOff | 1.000 | < 0.001 | 0.730 | < 0.001 |
| TBOn | 1.000 | < 0.001 | 0.714 | < 0.001 |
| TBMax | 1.000 | < 0.001 | 0.703 | < 0.001 |
| TBOff | 1.000 | < 0.001 | 0.721 | < 0.001 |

Note

ICC = Intraclass Correlation Coefficients.

TBOn = Displacement onset of tongue base retraction.

TBMax = Maximum displacement of tongue base retraction.

TBOff = Displacement offset of tongue base retraction.

HBOn = Displacement onset of hyoid bone excursion.

HBMax = Maximum displacement of hyoid bone excursion.

HBOff = Displacement offset of hyoid bone excursion.

## Adherence of paired-events to anticipated sequences

The following paired-event sequences were selected to investigate for their adherence and/or variability in swallowing trials:

1. Displacement onset of hyoid bone excursion always precedes displacement onset of tongue base retraction (*HBOn < TBOn*);

2. Displacement onset of hyoid bone excursion always precedes maximum displacement of tongue base retraction (*HBOn < TBMax*);

3. Displacement onset of tongue base retraction always precedes maximum displacement of hyoid bone excursion (*TBOn < HBMax*);

4. Maximum displacement of hyoid bone excursion always precedes maximum displacement of tongue base retraction (*HBMax < TBMax*);

5. Maximum displacement of tongue base retraction always precedes displacement offset of hyoid bone excursion (*TBMax < HBOff*);

6. Displacement offset of hyoid bone excursion always precedes displacement offset of tongue base retraction (*HBOff < TBOff*).

Results on percentage adherence to the anticipated paired-event sequences are summarized in Table 4. Adherence was defined as the number of swallows that follow the anticipated sequence, with a frame difference of at least one, out of the total number of swallows. Among the six paired-event sequences studied; two obligatory sequences, namely, *HBOn < TBMax* and *TBMax < HBOff* were found. The sequence *HBOn < TBOn* exhibited a high degree of adherence when considering all bolus types (98.1%) and individual bolus types (range = 97.4% - 100.0%). The *TBOn < HBMax* sequence also showed a high degree of adherence (i.e. > 90%), except for swallows of *10mL x thick* boluses. Nevertheless, noticeable variability was found in the *HBMax < TBMax* sequence (range of percentage adherence = 65.9% - 87.2%) and *HBOff < TBOff* sequence (range of percentage adherence = 75.6% - 82.9%). In all pairwise bolus type comparisons in level of adherence, the only significant difference was found

**Table 2. Mean time difference (in ms) between the onset of hyoid bone excursion (*HBOn*) and different events.**

| Bolus type | TBOn | HBMax | TBMax | HBOff | TBOff |
|---|---|---|---|---|---|
| All bolus types (n = 161) | 115.88 [60.07] | 237.19 [77.14] | 300.85 [65.53] | 555.51 [93.81] | 621.89 [117.04] |
| 5mL x thin (n = 41) | 118.90 [60.60] | 256.86 [85.16] | 310.21 [72.75] | 580.79 [99.31] | 626.52 [91.36] |
| 5mL x thick (n = 40) | 96.88 [44.65] | 225.00 [64.55] | 285.16 [46.51] | 524.22 [75.72] | 587.50 [100.13] |
| 10mL x thin (n = 41) | 128.05 [71.54] | 256.10 [84.50] | 313.26 [77.34] | 576.98 [102.01] | 658.54 [162.14] |
| 10mL x thick (n = 39) | 119.39 [57.74] | 209.13 [62.15] | 294.07 [58.56] | 538.46 [85.57] | 613.78 [90.25] |

Note

TBOn = Displacement onset of tongue base retraction.

TBMax = Maximum displacement of tongue base retraction.

TBOff = Displacement offset of tongue base retraction.

HBOn = Displacement onset of hyoid bone excursion.

HBMax = Maximum displacement of hyoid bone excursion.

HBOff = Displacement offset of hyoid bone excursion.

[] = Standard deviation.

between *5mL x thin* and *10mL x thick* boluses in the *HBMax < TBMax* sequence ($\chi^2$ = 5.020, $df$ = 1, $N$ = 80, $p$ = 0.025; phi = -0.250).

Fisher's exact tests were conducted to examine the relationship between bolus type and adherence to the anticipated sequences of paired-event except for the sequences *HBOn < TBMax* and *TBMax < HBOff*, since these sequences showed 100% adherence for all bolus types (see Table 3). Statistically non-significant results ($p \geq 0.153$) were found in all the remaining paired-event sequences. The correlations between bolus type and adherence were analysed using Cramer's V statistics. The correlation coefficients and correlation strengths are also summarized in Table 5.

## Discussion

The present study aimed to establish temporal sequences followed by swallowing events and investigate the variability of these temporal sequences in non-dysphagic individuals. The oro-pharyngeal swallowing events were related to tongue base retraction and hyoid bone excursion and were identified using ultrasound imaging.

From the 161 swallows executed by 41 healthy young adults of both genders, the following sequence was established:

1. displacement onset of hyoid bone excursion (*HBOn*) occurs prior to

2. displacement onset of tongue base retraction (*TBOn*) occurs prior to

3. maximum displacement of hyoid bone excursion (*HBMax*) occurs prior to

4. maximum displacement of tongue base retraction (*TBMax*) occurs prior to

5. displacement offset of hyoid bone excursion (*HBOff*) occurs prior to

6. displacement offset of tongue base retraction (*TBOff*).

However, variability was noted in 14.3% of all swallows. Variability ranged from 12.5% to 15.4% when different bolus types were taken into account. This suggests that despite a

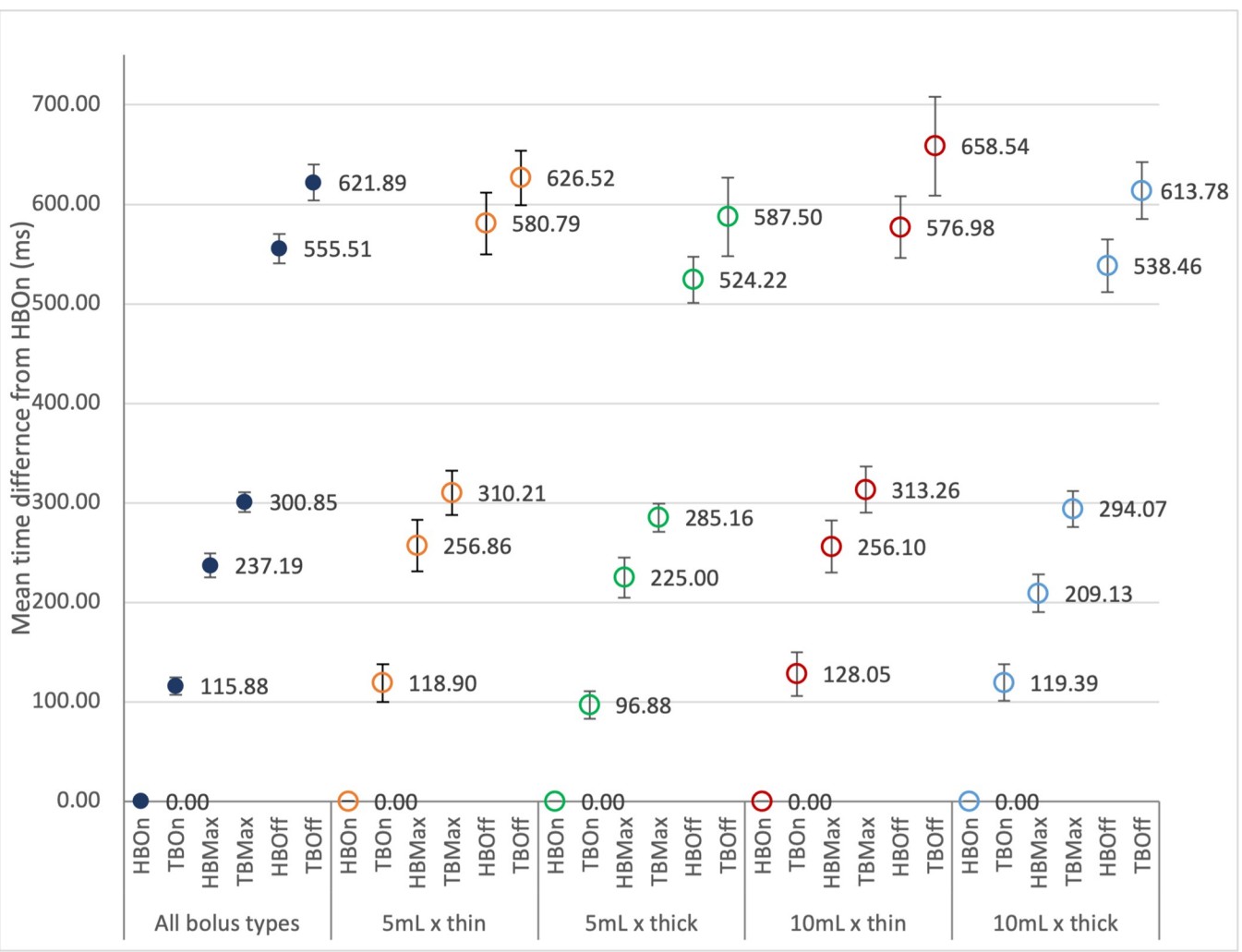

**Fig 4. Graphical presentation of mean time difference (in ms) from the onset of hyoid bone excursion (*HBOn*) for different events when swallowing different bolus types.** *Note*: TBOn = Displacement onset of tongue base retraction, TBMax = Maximum displacement of tongue base retraction, TBOff = Displacement offset of tongue base retraction, HBOn = Displacement onset of hyoid bone excursion, HBMax = Maximum displacement of hyoid bone excursion, HBOff = Displacement offset of hyoid bone excursion, Error bars represent 95% confidence intervals.

designated sequence us generally followed, individual differences are expected in non-eventful swallows of healthy adults. Variability in temporal measures of swallowing had been well documented in the review on 46 studies conducted by Molfenter and Steele [21].

Displacement onset of hyoid bone excursion (*HBOn*) was found to be the first event occurring in the majority of swallows. When the paired-event "displacement onset of hyoid bone excursion (i.e. the first event of the general sequence) always precedes displacement onset of tongue base retraction (i.e. the second event of the general sequence)" (*HBOn* < *TBOn*) was investigated, a high percentage of swallows adhered to this anticipated sequence. This is in accordance with the findings by Mendell and Logemann–the onset of hyoid bone elevation would precede the onset of posterior tongue base movement in swallows captured using videofluoroscopy regardless of the volume and consistency of bolus being studied [6]. Hyoid bone excursion, together with laryngeal excursion, plays a crucial role in ensuring swallowing safety. During the excursion, the larynx is elevated such that the airway closed off by the epiglottis and bolus is diverted away from the airway [29]. The excursion also leads the UES to open

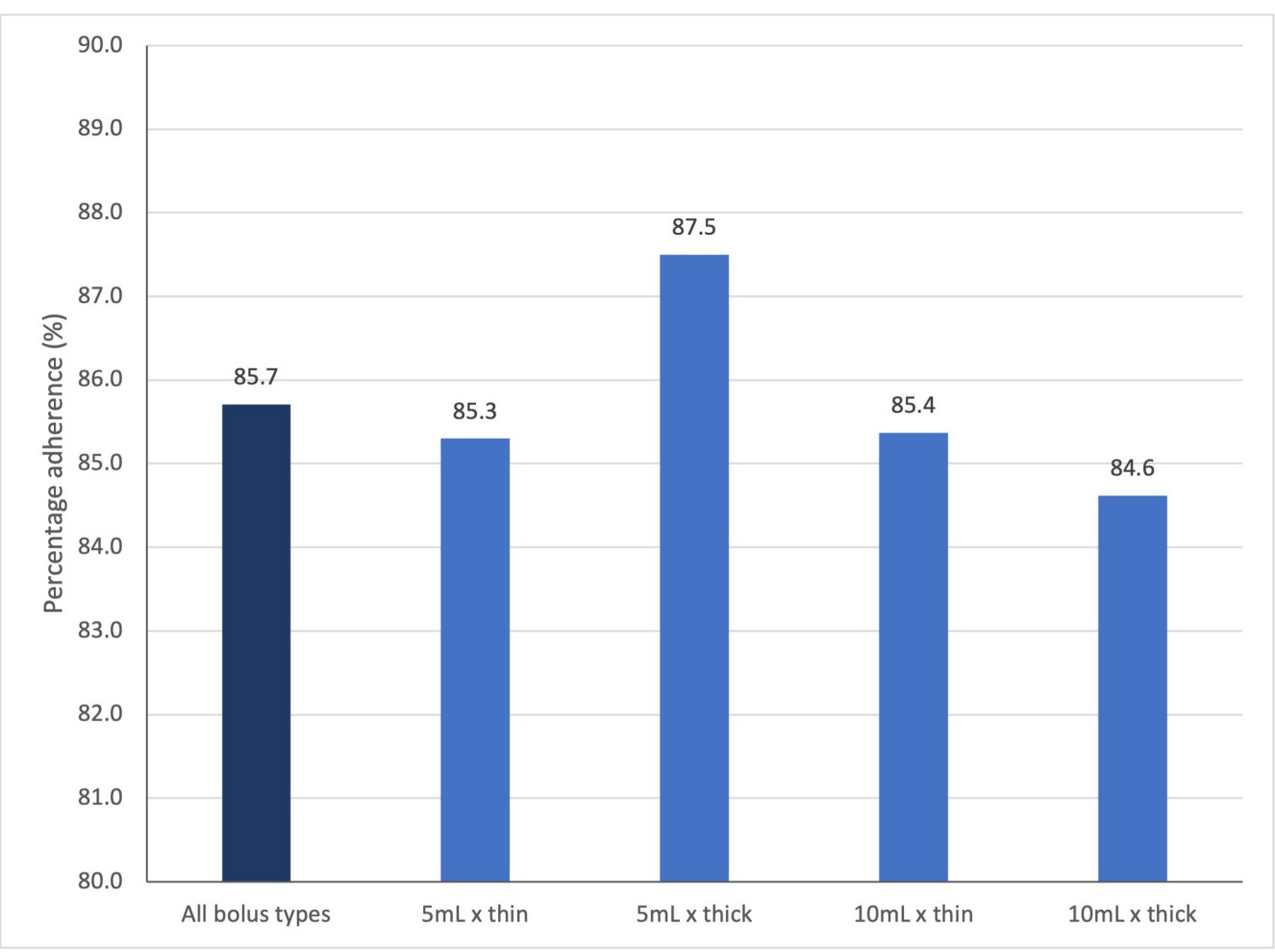

**Fig 5. Percentage adherence of trials that follows the designated temporal sequence of swallowing events.**

[29]. It is not surprising to find *HBOn* occurs even before the commencement of tongue base retraction, the movement that is responsible for bolus delivery to the pharynx [2], in non-dysphagic subjects with every little variability.

Significant main effects from the ANOVAs suggested that the timing at which *HBMax*, *TBMax*, *HBOff*, and *TBOff* occur may be influenced by bolus consistencies, with thick boluses resulting in earlier occurrence of these events than thin boluses. A previous study by Nagy et al. stated that thicker boluses would result in faster hyoid bone movements [23]. In the present study, this may be reflected by arriving at the maximum hyoid bone displacement at an earlier time point. Since the timing of swallowing events was measured by the time difference from a reference event (i.e. *HBOn*, the first-occurring event), the significant difference in the timing of *HBMax* itself may account for differences in timing for any subsequent events. Despite the main effect of bolus consistency was found in *TBMax*, *HBOff* and *TBOff*, it is inconclusive to claim that the significant differences in timing of these events were resulted from thicker boluses. On the other hand, the non-significant volume effect on the timing of hyoid bone-related events echoes with another study by Nagy and colleagues, which also found that difference in bolus volume did not necessarily lead to difference in hyoid bone displacement velocity [22]. The timing of *HBMax* and any subsequent relevant events may, therefore, not differ with bolus volume.

**Table 3. Results of the two-way ANOVAs on the time difference between the onset of hyoid bone excursion (*HBOn*) and different events.**

| Swallowing events and effects | df | F | p | $\eta_p^2$ |
|---|---|---|---|---|
| TBOn | | | | |
| Volume x Consistency[#] | 1 | 0.508 | 0.477 | 0.003 |
| Volume[†] | 1 | 2.847 | 0.094 | 0.018 |
| Consistency[†] | 1 | 2.674 | 0.104 | 0.017 |
| HBMax | | | | |
| Volume x Consistency[#] | 1 | 0.407 | 0.524 | 0.003 |
| Volume[†] | 1 | 0.493 | 0.483 | 0.003 |
| Consistency[†] | 1 | 11.087 | 0.001* | 0.066 |
| TBMax | | | | |
| Volume x Consistency[#] | 1 | 0.082 | 0.775 | 0.001 |
| Volume[†] | 1 | 0.340 | 0.561 | 0.002 |
| Consistency[†] | 1 | 4.645 | 0.033* | 0.029 |
| HBOff | | | | |
| Volume x Consistency[#] | 1 | 0.392 | 0.532 | 0.002 |
| Volume[†] | 1 | 0.131 | 0.718 | 0.001 |
| Consistency[†] | 1 | 10.880 | 0.001* | 0.065 |
| TBOff | | | | |
| Volume x Consistency[#] | 1 | 0.025 | 0.875 | < 0.001 |
| Volume[†] | 1 | 2.573 | 0.111 | 0.016 |
| Consistency[†] | 1 | 5.313 | 0.022* | 0.033 |

Note

[#]Interaction.

[†]Main effect.

*significant at 0.05 level.

TBOn = Displacement onset of tongue base retraction.

TBMax = Maximum displacement of tongue base retraction.

TBOff = Displacement offset of tongue base retraction.

HBOn = Displacement onset of hyoid bone excursion.

HBMax = Maximum displacement of hyoid bone excursion.

HBOff = Displacement offset of hyoid bone excursion.

Maximum displacement and displacement offset of swallowing movements are less discussed in the literature, including those related to tongue base retraction and hyoid bone excursion. Among all the paired-events being studied, two obligatory sequences were found in the present study, they are, the *HBOn < TBMax* and *TBMax < HBOff*. Although the percentage adherence was close to 100%, the sequence of the paired-event *HBOn < TBOn* was yet to be considered obligatory. The sequence related to hyoid bone displacement onset and maximum tongue base retraction was further examined to better understand the temporal relationship between the two movements. It was found that *HBOn* may not always precede *TBOn* but *TBMax*, indicating that airway protection and UES opening always commence before the maximum bolus propulsion towards the pharynx in healthy subjects. Another obligatory sequence was found in the paired-event *TBMax < HBOff*. The difference in time taken from *HBOn* to *TBMax* and to *HBOff* ranged from 239.06ms to 270.58ms in different bolus types (see Table 2 and Fig 4). On top of propelling the bolus towards the pharynx, tongue base retraction is considered to associate with valleculae clearance during swallows [30], especially when it is maximally retracted. Maintaining maximum airway protection and maximum UES opening until

**Table 4. Percentage adherence to the anticipated sequence of paired-events.**

| Bolus type | HBOn < TBOn | HBOn < TBMax | TBOn < HBMax | HBMax < TBMax | TBMax < HBOff | HBOff < TBOff |
|---|---|---|---|---|---|---|
| All bolus types (n = 161) | 98.1 | 100.0 | 90.7 | 75.8 | 100.0 | 78.8 |
| 5mL x thin (n = 41) | 97.6 | 100.0 | 92.7 | 65.9 | 100.0 | 75.6 |
| 5mL x thick (n = 40) | 97.5 | 100.0 | 92.5 | 77.5 | 100.0 | 77.5 |
| 10mL x thin (n = 41) | 100.0 | 100.0 | 92.7 | 73.2 | 100.0 | 82.9 |
| 10mL x thick (n = 39) | 97.4 | 100.0 | 84.6 | 87.2 | 100.0 | 79.5 |

Note

TBOn = Displacement onset of tongue base retraction.

TBMax = Maximum displacement of tongue base retraction.

TBOff = Displacement offset of tongue base retraction.

HBOn = Displacement onset of hyoid bone excursion.

HBMax = Maximum displacement of hyoid bone excursion.

HBOff = Displacement offset of hyoid bone excursion.

the tongue base is retracted maximally for at least 239.06ms may ensure swallowing safety to the greatest extent.

The paired-event *TBOn < HBMax* showed > 90% adherence for most bolus types. In general, tongue base retraction commences before maximum hyoid bone excursion is reached. However, for a challenging bolus type like *10mL x thick*, maximum hyoid bone excursion may be achieved early in some subjects. This is in line with the findings suggested by Nagy and colleagues [23], which stated that velocity of hyoid bone movement increases with bolus consistency in non-dysphagic subjects. The paired-events *HBMax < TBMax* and *HBOff < TBOff* have relatively low levels of adherence. Individual variability is allowed in these event sequences without compromising swallowing safety.

**Table 5. Results of Fisher's exact and Cramer's V tests on the association between bolus type and adherence to the anticipated sequence of paired-events.**

| Paired-events | *p* value of Fisher's exact test | Cramer's V | Strength[2] |
|---|---|---|---|
| HBOn < TBOn | 0.805 | 0.081 | Weak |
| HBOn < TBMax[1] | -- | -- | -- |
| TBOn < HBMax | 0.589 | 0.118 | Moderate |
| HBMax < TBMax | 0.153 | 0.179 | Strong |
| TBMax < HBOff[1] | -- | -- | -- |
| HBOff < TBOff | 0.874 | 0.067 | Weak |

Note

TBOn = Displacement onset of tongue base retraction.

TBMax = Maximum displacement of tongue base retraction.

TBOff = Displacement offset of tongue base retraction.

HBOn = Displacement onset of hyoid bone excursion.

HBMax = Maximum displacement of hyoid bone excursion.

HBOff = Displacement offset of hyoid bone excursion.

[1] Fisher's exact test was not computed.

[2] Association strength based on Akoglu [28].

It is interesting to note that from the largely non-significant pairwise comparisons, bolus type had little or no effect on the adherence to the anticipated sequences of paired-events. Statistical analyses on the relationship between bolus type and adherence suggested that the two variables are not associated with each other in all paired-events studied. Examinations on the strength of relationship, nevertheless, shows that bolus types might have moderate and strong associations with adherence for the paired-events *TBOn < HBMax* and *HBMax < TBMax* respectively in healthy adults. The level of adherence for *TBOn < HBMax* was noticeably lower when the subjects swallowed the *10mL x thick* boluses (see Table 4). For *HBMax < TBMax*, the more challenging the bolus (i.e. *10mL x thick*), the higher was the level of adherence. These findings suggest that besides individual variability, bolus property may also have an effect on adherence to anticipated sequence; especially when these two paired-events are taken into account. Two published systematic reviews also suggested that bolus properties like of volume and density could be potential sources of temporal [21] and spatial [31] variability of swallowing. Bolus types varying in different dimensions should be taken into consideration in future studies on swallowing kinematics.

## Limitations and recommendations

Considering the substantial amount of time and labour intensiveness required for data extraction, data were extracted from only one trial for each bolus type in each subject. It is recommended that data may be obtained from more trials to eliminate possible measurement and/or extraction errors, and to examine the extent of possible within-subject between-trial variability. Inter-examiner reliability was moderate. Measures like consensus meeting between examiners may be taken to enhance reliability. It is also recommended to expand the investigations to swallowing movements other than tongue base retraction and hyoid bone excursion. Besides, on top of the temporal aspects; spatial aspects (e.g. displacement) of swallowing events may be investigated in future studies. Despite that duration of movements may be deduced from the differences in timing of events (e.g. the duration of maintaining maximum tongue base retraction may be deduced from the difference between *TBMax* and *TBOff*), analyses on movement duration and/or latency among events may be also be included in future studies.

## Conclusion

Ultrasound imaging is a non-invasive technique, both physically and radioactively, that can be applied to individuals at different ages and with different backgrounds. It is also considered more accessible, as compared to traditional swallowing examination techniques like videofluoroscopy and FEES. The present study has further provided evidence of the utility of ultrasound imaging for the identification of swallowing related structures (i.e. tongue base and hyoid bone) and movements (tongue base retraction and hyoid bone excursion). Two obligatory sequences and the timing of swallowing events related to the essential movements were found and individual variability was also observed in the measurements of the movements among healthy adults. Further, it is found that bolus property may have an effect on the timing and sequence of swallowing events. Findings of the present study lay the groundwork for future studies that investigate and compare temporal swallowing data between healthy and dysphagic individuals sonographically, and provide further support for adopting ultrasound imaging in the examination and diagnosis of swallowing in clinical settings.

## Supporting information

**S1 Data.**
(XLSX)

## Acknowledgments

The authors would like to thank all participants in the study and Ms Katrina Ng for her assistance in data collection.

## Author Contributions

**Conceptualization:** Elaine Kwong, Phoebe Tsz-Ching Shek, Man-Tak Leung, Yong-Ping Zheng.

**Data curation:** Elaine Kwong, Phoebe Tsz-Ching Shek, Yong-Ping Zheng.

**Formal analysis:** Elaine Kwong, Phoebe Tsz-Ching Shek, Wilson Yiu Shun Lam.

**Funding acquisition:** Elaine Kwong, Man-Tak Leung, Yong-Ping Zheng.

**Investigation:** Elaine Kwong, Man-Tak Leung, Yong-Ping Zheng, Wilson Yiu Shun Lam.

**Methodology:** Elaine Kwong, Phoebe Tsz-Ching Shek, Man-Tak Leung, Yong-Ping Zheng.

**Project administration:** Elaine Kwong, Phoebe Tsz-Ching Shek, Yong-Ping Zheng.

**Supervision:** Elaine Kwong, Man-Tak Leung, Yong-Ping Zheng.

**Validation:** Elaine Kwong.

**Writing – original draft:** Elaine Kwong, Phoebe Tsz-Ching Shek, Man-Tak Leung, Yong-Ping Zheng.

**Writing – review & editing:** Elaine Kwong, Wilson Yiu Shun Lam.

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
