## [Decision Letter · Decision Letter 0]

11 Jan 2022

PONE-D-21-34358Temporal measures of oropharyngeal swallowing events identified using ultrasound imaging in healthy young adultsPLOS ONE

Dear Dr. Kwong,

Thank you for submitting your manuscript to PLOS ONE. After careful consideration, we feel that it has merit but does not fully meet PLOS ONE’s publication criteria as it currently stands. Therefore, we invite you to submit a revised version of the manuscript that addresses the points raised during the review process.

We look forward to receiving your revised manuscript.

Kind regards,

Hyojung Choo

Academic Editor

PLOS ONE

Journal Requirements:

Reviewers' comments:

Reviewer's Responses to Questions

**Comments to the Author**

1. Is the manuscript technically sound, and do the data support the conclusions?

Reviewer #1: Yes

Reviewer #2: No

2. Has the statistical analysis been performed appropriately and rigorously? 

Reviewer #1: Yes

Reviewer #2: Yes

3. Have the authors made all data underlying the findings in their manuscript fully available?

Reviewer #1: Yes

Reviewer #2: No

4. Is the manuscript presented in an intelligible fashion and written in standard English?

Reviewer #1: Yes

Reviewer #2: Yes

5. Review Comments to the Author

Reviewer #1: The study is interesting as well as the well-executed statistical analysis. However, to improve the quality of the manuscript I recommend to insert these following references:

- In the introduction emphasizes the role of Patient-reported measures in dysphagia as illustrated in an interesting systematic review indicated: Patel DA, Sharda R, Hovis KL, Nichols EE, Sathe N, Penson DF, Feurer ID, McPheeters ML, Vaezi MF, Francis DO. Patient-reported outcome measures in dysphagia: a systematic review of instrument development and validation. Dis Esophagus. 2017 May 1;30(5):1-23. doi: 10.1093/dote/dow028.

- In the introduction to suggest other methods of evaluating swallowing movement using ultrasonography: Matsuo T, Mstsuyama M, Nakatani K, Naoe M. Evaluation of swallowing movement using ultrasonography. Radiological Physics and Technology .2019 May; 13(1):62-68. doi:10.1007/s12194-019-00547-1

- In the discussion add a systematic review of hyoid and laryngeal movements as indicated in the following reference:

Sonja M. Molfenter, Catriona M. Steele. Physiological variability in the deglutition literature: hyoid and laryngeal kinematics. Dysphagia. 2011 Mar;26(1):67-74. doi: 10.1007/s00455-010-9309-x.

Reviewer #2: The study invetigated the time sequence of hyoid bone and tongue base movement by ultrasound when swallowing different volume and consistency of bolus. Obligatory and non-obligator sequence was identified and variability of adherence to several sequence was reported.

The topic is of clinical significance. However, several issues should be addressed and clarified to validate the study.

My comments are as follows:

1. The procedure of ultrasound exam should be specified in detail, such as the head support, placement and fixation of gel pad, transducer holding, because the stabilization of transducer and image quality is critical for accurate analysis of hyoid and tongue movemnent.

2.US is highly operator dependent. Is the operator experienced with US exam? The inadequate reliability of US would make the results doutful.

3. Are the subjects holding the bolus at mouth floor or in front of the tongue? This may significantly alter the results.

4.The localization of hyoid and tongue base in sonographic view should be elaborated.

In figure 1, the B indicate the acoustic shadow but not hyoid bone. The presence of acoustic shadow is extremely variable depending on the placement position and angle of transducer, therefore, should not be used for quantitative analysis. The margin of hyoid bone or the interface between hyoid bone/geniohyoid bone should be used as a consistent marker.

5. A clear definition of event, such as onset and offset of their movement is required.

6. Several previous studies have investigated the time sequence of swallowing events in relation to bolus consistency and volume, including some ultrasound studies. The introduction should briefly review these studies and point out what is lacking.

7. As the authors pointed out in the limitations, only one trial of swallow is analysed. This would significantly affect the validity of the study as large between-trial variation exists even in the same individual.

8. Table 2 & 3: are there any differences betwen groups?

9. Table 4. The effect of bolus volume and consistency should be considered separately.

10. The discussion should include more comparison with previous studies, instead of repeat of results. For example, how bolus consistenct affect movement sequence of hyoid bone and tongue base.

6. PLOS authors have the option to publish the peer review history of their article (what does this mean?). If published, this will include your full peer review and any attached files.

Reviewer #1: No

Reviewer #2: No

---

## [Author Response · Author response to Decision Letter 0]

2 Feb 2022

Please refer to the point-by-point responses in the attachment.

---

## [Decision Letter · Decision Letter 1]

21 Feb 2022

PONE-D-21-34358R1

Temporal measures of oropharyngeal swallowing events identified using ultrasound imaging in healthy young adults

PLOS ONE

Dear Dr. Kwong,

Thank you for submitting your manuscript to PLOS ONE. After careful consideration, we have decided that your manuscript does not meet our criteria for publication and must therefore be rejected.

Although we appreciate substantial improvement by revision, one of reviewers still found the fundamental issue of acquisition and analysis of ultrasonographic images that remained unsolved. The reviewer mentioned that "the presence of acoustic shadow is extremely variable depending on the placement position and angle of transducer, therefore, should not be used for quantitative analysis". This issues may reduce reproducibility of your research. Please consider to reanalyzing data with a reliable marker for ultrasound image analysis.

I am sorry that we cannot be more positive on this occasion, but hope that you appreciate the reasons for this decision.

Yours sincerely,

Hyojung Choo

Academic Editor

PLOS ONE

Additional Editor Comments:

We appreciate substantial improvement by revision. However, one of reviewers still found the fundamental issue of acquisition and analysis of ultrasonographic images that remained unsolved. The reviewer mentioned that "the presence of acoustic shadow is extremely variable depending on the placement position and angle of transducer, therefore, should not be used for quantitative analysis". This issues may reduce reproducibility of your research. Please consider to reanalyzing data with a reliable marker for ultrasound image analysis.

Reviewers' comments:

Reviewer's Responses to Questions

**Comments to the Author**

1. If the authors have adequately addressed your comments raised in a previous round of review and you feel that this manuscript is now acceptable for publication, you may indicate that here to bypass the “Comments to the Author” section, enter your conflict of interest statement in the “Confidential to Editor” section, and submit your "Accept" recommendation.

Reviewer #1: (No Response)

Reviewer #2: (No Response)

2. Is the manuscript technically sound, and do the data support the conclusions?

Reviewer #1: Yes

Reviewer #2: No

3. Has the statistical analysis been performed appropriately and rigorously? 

Reviewer #1: Yes

Reviewer #2: Yes

4. Have the authors made all data underlying the findings in their manuscript fully available?

Reviewer #1: Yes

Reviewer #2: Yes

5. Is the manuscript presented in an intelligible fashion and written in standard English?

Reviewer #1: Yes

Reviewer #2: Yes

6. Review Comments to the Author

Reviewer #1: The authors have well modify the manuscript. We encourage you to accept it in its current form, which is a very interesting study.

Reviewer #2: Thank you for taking into account the issues raised. The authors have made substantial revision, however, the foundamental issue of acuisition and analysis of ultrasonographic images remained unsolved. The figure is re-annotated according to suggestions but the analsis was based on previous anotation, using acoustic shadow of hyoid bone as marker. As mentioned in previous comments, the presence of acoustic shadow is extremely variable depending on the

placement position and angle of transducer, therefore, should not be used for quantitative analysis.

7. PLOS authors have the option to publish the peer review history of their article (what does this mean?). If published, this will include your full peer review and any attached files.

Reviewer #1: No

Reviewer #2: No

- - - - -

---

## [Author Response · Author response to Decision Letter 1]

11 Mar 2022

Please refer to the appeal letter (cover letter).

---

## [Decision Letter · Decision Letter 2]

18 Apr 2022

PONE-D-21-34358R2Temporal measures of oropharyngeal swallowing events identified using ultrasound imaging in healthy young adultsPLOS ONE

Dear Dr. Kwong,

Thank you for submitting your manuscript to PLOS ONE. After careful consideration, we feel that it has merit but does not fully meet PLOS ONE’s publication criteria as it currently stands. Therefore, we invite you to submit a revised version of the manuscript that addresses the points raised during the review process.

ACADEMIC EDITOR: Please insert comments here and delete this placeholder text when finished. Be sure to:Indicate which changes you require for acceptance versus which changes you recommendAddress any conflicts between the reviews so that it's clear which advice the authors should followProvide specific feedback from your evaluation of the manuscriptPlease ensure that your decision is justified on PLOS ONE’s publication criteria and not, for example, on novelty or perceived impact.

We look forward to receiving your revised manuscript.

Kind regards,

Hyojung Choo

Academic Editor

PLOS ONE

Journal Requirements:

Additional Editor Comments (if provided):

Reviewers' comments:

Reviewer's Responses to Questions

**Comments to the Author**

1. If the authors have adequately addressed your comments raised in a previous round of review and you feel that this manuscript is now acceptable for publication, you may indicate that here to bypass the “Comments to the Author” section, enter your conflict of interest statement in the “Confidential to Editor” section, and submit your "Accept" recommendation.

Reviewer #3: (No Response)

2. Is the manuscript technically sound, and do the data support the conclusions?

Reviewer #3: Partly

3. Has the statistical analysis been performed appropriately and rigorously? 

Reviewer #3: Yes

4. Have the authors made all data underlying the findings in their manuscript fully available?

Reviewer #3: Yes

5. Is the manuscript presented in an intelligible fashion and written in standard English?

Reviewer #3: Yes

6. Review Comments to the Author

Reviewer #3: The present study was designed to establish the temporal property of swallowing events

using ultrasound imaging and to investigate the variability of the sequences and effect of bolus type on it in healthy individuals. The study on ultrasound imaging is interesting, but some concerns are raised mainly with respect to the methods.

Comment 1

P3, L8

What is “four stages”? The authors are recommended to explain this term in brief because not all readers are familiar with swallowing physiology.

Comment 2

P3, L6 from the bottom, Patients with radiotherapy

Do the authors mean “patients with head and neck cancer”?

Comment 3

Materials and methods, Procedure

P6, Materials and equipment

How was the sampling rate of recordings?

Comment 4

P8, L2

The authors selected “the best image” of the three data. How did the authors determine the best? In addition, were there any variation of time sequence across the three images? This may be helpful to interpret reproducibility and reliability.

Comment 5

P8, L7

In Fig. 1, the authors depicted the tongue base by green line, but the real line of tongue surface was not clearly shown. Further, the authors are recommended to show all the pictures at time of TBOn, TBMax and TBOff in one trial.

P20, Limitations

One of the biggest limitations is that the authors measured only the time of onset, peak (maximum) and offset of TB and HB. If was the movement just straight? How was the time duration of the tongue base staying at the maximum?

7. PLOS authors have the option to publish the peer review history of their article (what does this mean?). If published, this will include your full peer review and any attached files.

Reviewer #3: No

---

## [Decision Letter · Decision Letter 3]

16 Jun 2022

Temporal measures of oropharyngeal swallowing events identified using ultrasound imaging in healthy young adults

PONE-D-21-34358R3

Dear Dr. Kwong,

We’re pleased to inform you that your manuscript has been judged scientifically suitable for publication and will be formally accepted for publication once it meets all outstanding technical requirements.

Kind regards,

Hyojung Choo

Academic Editor

PLOS ONE

Additional Editor Comments (optional):

Reviewers' comments:

Reviewer's Responses to Questions

**Comments to the Author**

1. If the authors have adequately addressed your comments raised in a previous round of review and you feel that this manuscript is now acceptable for publication, you may indicate that here to bypass the “Comments to the Author” section, enter your conflict of interest statement in the “Confidential to Editor” section, and submit your "Accept" recommendation.

Reviewer #3: All comments have been addressed

2. Is the manuscript technically sound, and do the data support the conclusions?

Reviewer #3: Yes

3. Has the statistical analysis been performed appropriately and rigorously? 

Reviewer #3: Yes

4. Have the authors made all data underlying the findings in their manuscript fully available?

Reviewer #3: Yes

5. Is the manuscript presented in an intelligible fashion and written in standard English?

Reviewer #3: Yes

6. Review Comments to the Author

Reviewer #3: (No Response)

7. PLOS authors have the option to publish the peer review history of their article (what does this mean?). If published, this will include your full peer review and any attached files.

Reviewer #3: **Yes: **Makoto Inoue

---

## [Editor Report · Acceptance letter]

20 Jun 2022

PONE-D-21-34358R3 

Temporal measures of oropharyngeal swallowing events identified using ultrasound imaging in healthy young adults 

Dear Dr. Kwong:

I'm pleased to inform you that your manuscript has been deemed suitable for publication in PLOS ONE. Congratulations! Your manuscript is now with our production department. 

Kind regards, 

on behalf of

Dr. Hyojung Choo 

Academic Editor

PLOS ONE